# A transcriptome-wide association study of uterine fibroids to identify potential genetic markers and toxic chemicals

**Gayeon Kim**[1]☯, **Gyuyeon Jang**[1]☯, **Jaeseung Song**[1], **Daeun Kim**[1], **Sora Lee**[1], **Jong Wha J. Joo**[2], **Wonhee Jang**[1]*

**1** Department of Life Sciences, Dongguk University-Seoul, Seoul, Republic of Korea, **2** Department of Computer Science and Engineering, Dongguk University-Seoul, Seoul, South Korea

☯ These authors contributed equally to this work.
* wany@dongguk.edu

**Data Availability Statement:** All results are available as Supporting Information files, and the source files can be obtained from the TWAS/

## Abstract

Uterine fibroid is one of the most prevalent benign tumors in women, with high socioeconomic costs. Although genome-wide association studies (GWAS) have identified several loci associated with uterine fibroid risks, they could not successfully interpret the biological effects of genomic variants at the gene expression levels. To prioritize uterine fibroid susceptibility genes that are biologically interpretable, we conducted a transcriptome-wide association study (TWAS) by integrating GWAS data of uterine fibroid and expression quantitative loci data. We identified nine significant TWAS genes including two novel genes, *RP11-282O18.3* and *KBTBD7*, which may be causal genes for uterine fibroid. We conducted functional enrichment network analyses using the TWAS results to investigate the biological pathways in which the overall TWAS genes were involved. The results demonstrated the immune system process to be a key pathway in uterine fibroid pathogenesis. Finally, we carried out chemical–gene interaction analyses using the TWAS results and the comparative toxicogenomics database to determine the potential risk chemicals for uterine fibroid. We identified five toxic chemicals that were significantly associated with uterine fibroid TWAS genes, suggesting that they may be implicated in the pathogenesis of uterine fibroid. In this study, we performed an integrative analysis covering the broad application of bioinformatics approaches. Our study may provide a deeper understanding of uterine fibroid etiologies and informative notifications about potential risk chemicals for uterine fibroid.

## Introduction

Uterine fibroids (UFs) or uterine leiomyoma are the most common benign tumors in women, and >40% of Caucasian women will be diagnosed with UF at least once in their lifetime [1]. Thirty percent of UF patients have severe symptoms such as uterine bleeding, pelvic pain, and infertility [2]. Reportedly, UF can transform into malignant tumors in some patients [3, 4].

FUSION webpage (http://gusevlab.org/projects/fusion/) which is freely accessible.

**Funding:** This work was supported by the National Research Foundation of Korea (NRF) grant funded by the Korea government (MSIT) (No. NRF-2021R1A2C1008804). This work was supported by the Dongguk University Research Fund of 2020. The funders had no role in study design, data collection and analysis, decision to publish, or preparation of the manuscript.

**Competing interests:** The authors have declared that no competing interests exist.

The high prevalence of UF has a serious impact on annual healthcare costs all over the world; however, the pathogenesis of UF has not been completely understood [2].

Previous studies reported several genetic risk factors affecting UF [5–9]. A Finnish twin cohort study identified the strong heritability of UF, estimating that monozygotic twins had twice the incidence rate of UF compared with dizygotic twins [5]. Chromosomal abnormalities such as trisomy 12 and rearrangements of chromosomes 12, 13, and/or X are involved in the growth of UF [6]. The overexpression of *High mobility group AT-hook 2*, located in 12q 14–15, is related to the development of UF with or without chromosomal rearrangements [7]. Germ-line mutations in *fumarate hydratase*, a tumor suppressor gene, stimulate benign and malignant tumor development of UF [8, 9]. Deletion of *collagen type* IV *alpha 5* and *6 chain* mapped to chromosome X is also known for its association with UF [10].

Several pre-existing conditions and environmental substances such as sex steroids, obesity, hypertension, and endocrine-disrupting chemicals (EDCs) are reported to stimulate the pathogenesis and growth of UF [11–14]. Steroid hormones are involved in the proliferation and differentiation of uterine cells; in particular, estrogen and progesterone are regarded as agonists of fibroid growth [15, 16]. A previous study reported that obese women have a high prevalence of UF and body mass index levels have a strong positive correlation with fibroid growth [17]. Faerstein *et al.* identified that hypertension patients have high risk of UF pathogenesis [18]. Exposure to EDCs can also increase the risk of UF onset [19]. Despite the biological effects of EDCs, there have been very few studies on how these chemicals regulate biological responses to affect the pathogenesis of UF.

Recent genome-wide association studies (GWAS) identified several risk loci attributed to the development of UF. Nakamura's group proposed 10q24.33, 11p15.5, and 22q13.1 regions as risk loci from their Japanese cohort, and Zhang *et al.* reported 2q32.2 and 1q42.2 regions as risk loci using African-American and European-American populations [20, 21]. A GWAS meta-analysis on UF for European ancestry conducted by Gallagher's group reported eight risk loci: 2p23.2, 4q22.3, 6p21.31, 7q31.2, 10p11.22, 11p14.1, 12q15, and 12q24.31 [22]. Even though GWAS discovers risk loci via the association between single nucleotide polymorphism (SNP) and the disease, it is hard to determine what genetic effects are derived from the expression levels of risk variants. Transcriptome-wide association study (TWAS) is a useful solution to overcome such limitations by integrating GWAS with expression quantitative trait loci (eQTLs). TWAS utilizes pre-computed predictive models of gene expression trained by reference eQTL data to impute gene expression from large-scale genotype data [23]; it prioritizes putative causal genes where the *cis*-genetic component is associated with the disease [24].

Herein, we conducted a TWAS by using the meta-analyzed GWAS summary statistics data of UF with eQTL weight panels derived from large-scale consortium data and reference linkage disequilibrium (LD) matrix [22, 23, 25–27]. Conditional and joint analyses were performed to demonstrate the expressional independence of and associations in the TWAS and genes, and then carried out gene set enrichment analysis (GSEA) to explore biological functions. Finally, we conducted a chemical–gene interaction analysis using the comparative toxicogenomics database (CTD) to identify toxic chemicals associated with the expression of significant UF TWAS genes. We believe that our results may provide new insights into the pathogenesis of UF and useful information on how toxic chemicals affect the development of UF.

## Methods

### Data collection

The GWAS summary statistics data (GCST009158) were retrieved from the GWAS Catalog (https://www.ebi.ac.uk/gwas/studies/GCST009158). The GCST009158 dataset consists of

20,406 UF patients and 223,918 control subjects of European ancestry and is currently the largest publicly available UF GWAS data. Eight tissue-specific eQTL reference panels related to UF and reference LD data from the 1000 Genomes Project were retrieved from the functional summary-based imputation (FUSION) webpage (http://gusevlab.org/projects/fusion/) [24, 27]. Six tissue panels from Genotype-Tissue Expression version 7 (GTEx v7) including three female reproductive organs (ovary, uterus, and vagina), two tissue panels related to the regulation of ovarian hormones (hypothalamus and pituitary), and whole blood panel were selected as reference panels [23]. To uncover as many associations as possible, two different blood panels from individual studies representing whole blood (Young Finns Study, YFS) and peripheral blood (Netherlands Twin Register, NTR) were also included following previous studies [24, 25, 28].

## Transcriptome-wide association study

A TWAS was performed using the FUSION tool with default settings [24]. The GWAS summary statistics data file was converted into a sumstats-formatted file prior to transcriptomic imputation (TI) using the LD-score regression (LDSC), and the results from major histocompatibility complex regions were excluded to prevent inflating the association statistics [29]. TI was conducted using eight tissue-specific eQTL reference panels and the LD reference data from the European population with FUSION. To obtain statistically robust signatures, Bonferroni-corrected thresholds were used as significance thresholds to identify transcriptome-wide significant associations ($P < 0.05$/sum of SNP–gene pairs across the tissue panels (26,279) = $\sim 1.90 \times 10^{-6}$).

The GWAS summary statistics data were analyzed by the functional mapping and annotation (FUMA) for comparison with the TWAS of FUSION [30]. Two gene-prioritizing tests of FUMA—SNP2GENE process and multi-marker analysis of genomic annotation (MAGMA)—were conducted as post-GWAS annotation analyses. The SNP2GENE process maps SNPs to neighboring genes based on the physical position, eQTL associations, and chromatin interaction information [30]. MAGMA, which is one of the most widely used post-GWAS annotations, prioritizes genes associated with SNPs based on a multiple regression model [31]. Moreover, we performed colocalization tests with the COLOC R package to determine whether the gene expression signals were colocalized with the GWAS signals [32]. COLOC was conducted with FUSION software, and five posterior probabilities (PP0–4) were calculated corresponding to five different hypotheses (H0–4). H0: no association; H1: functional association but no GWAS association; H2: GWAS association but no functional association; H3: association with gene expression and GWAS signals but each is independent; H4: gene expression and GWAS association are colocalized. Genes satisfying the threshold of PP3+-PP4 > 0.8 and PP4/PP3 > 2 were prioritized in the colocalization tests, following previous studies [28, 33].

## Conditional and joint analyses

Conditional and joint analyses were conducted to identify independent TWAS genes in a specific locus harboring multiple TWAS associations after conditioning on the expression of TWAS genes using the FUSION.post_process.R code provided by FUSION. TWAS associations that were statistically significant after Bonferroni-correction were subjected to the conditional and joint analyses. Jointly significant genes in a locus were regarded as the robust genetic signatures for UF.

## Functional network analysis of TWAS results

To interpret the systemic biological roles of TWAS genes, GSEA was conducted with the Metascape, a web-based gene list annotation tool [34]. In order to analyze the broad genetic

signatures of UF, GSEA was conducted with marginally significant TWAS associations ($P < 0.05$) instead of the Bonferroni-adjusted threshold ($P < 1.90 \times 10^{-6}$). For eight tissues, individual lists of genes positively associated with UF risks (TWAS Z-score $> 0$) were analyzed using reference gene sets provided by the Metascape tool to annotate its biological function. The identical process was applied to genes negatively associated with UF risks (TWAS Z-score $< 0$). These analyses were conducted with default settings for the Metascape and the results were retrieved in Cytoscape file format for further analyses.

To detect representative pathways that may play crucial roles in the pathogenesis of UF, functional enrichment network analysis was conducted on the enriched biological pathways identified by the Metascape ($P < 0.01$). The functional enrichment networks comprising the pathways identified by the GSEA with TWAS genes were visualized using Cytoscape (v. 3.8.2) [35]. Duplicated enriched pathways were removed before the construction of functional enrichment networks. Among the networks, sub-network clusters consisting of closely inter-connected pathways were identified by molecular complex detection (MCODE) [36].

### Chemical–gene interaction analysis

To identify the chemical risk factors of UF, chemical–gene interaction analysis was conducted by CTD using the significant TWAS genes ($P < 1.90 \times 10^{-6}$). CTD provides curated information on chemical–gene/protein interactions, chemical–disease relationships, and gene–disease relationships from peer-reviewed scientific literature [37]. The analysis was performed by setting the organism as *Homo sapiens*. To obtain chemicals that may increase the risk of UF onset, significant TWAS genes with positive or negative Z-scores were respectively provided as input gene sets into the CTD. The chemicals were estimated to increase the expression levels of significant TWAS genes with positive Z-scores (TWAS Z-score $> 0$) or to decrease those with negative Z-scores (TWAS Z-score $< 0$). In short, chemicals expected to be involved in the pathogenesis of UF, were selected as potentially toxic chemicals for UF.

## Results

### Prioritization of susceptibility genes for UF using TWAS

To identify risk genes significantly associated with the pathogenesis of UF, we conducted a TWAS using currently the largest GWAS summary statistics dataset (GCST009158; total: 244,324; number of UF patients: 20,406, number of controls: 223,918) of European UF and eight eQTL tissue panels with the FUSION tool. S1 Table lists all 26,279 TWAS associations. The result showed 10 significant TWAS associations between the predicted expression of eQTL panels and UF, identifying nine genes as risk genes for UF after Bonferroni-correction ($P < 1.90 \times 10^{-6}$) (Figs 1 and S1). The nine significant TWAS genes were *secretoglobin family 1C member 1* (LOC653486), *proteasome 26S subunit, non-ATPase 13* (PSMD13), *RP11-282O18.3, M-phase phosphoprotein 9* (MPHOSPH9), *strawberry notch homolog 1* (SBNO1), *ADP ribosylation factor like GTPase 6 interacting protein 4* (ARL6IP4), *SET domain containing (lysine methyltransferase) 8* (SETD8), *kelch repeat and bric-a-brac, tramtrack, and broad-complex domain containing 7* (KBTBD7), and *mitochondrial ribosomal protein S31* (MRPS31) (Table 1). Among these nine significant genes, eight genes were detected in one of the three different blood panels (GTEx whole blood, NTR, or YFS), and only *RP11-282O18.3* showed a significant association in the uterus panel. While significant GWAS signals were observed on most chromosomes, the significant TWAS genes were only detected in chromosomes 11, 12, and 13 (S2A Fig). These results may be based on the cytogenetic rearrangement of the specific loci (12q15, 12q24, and 13q) that are characteristics of UF [38–40].

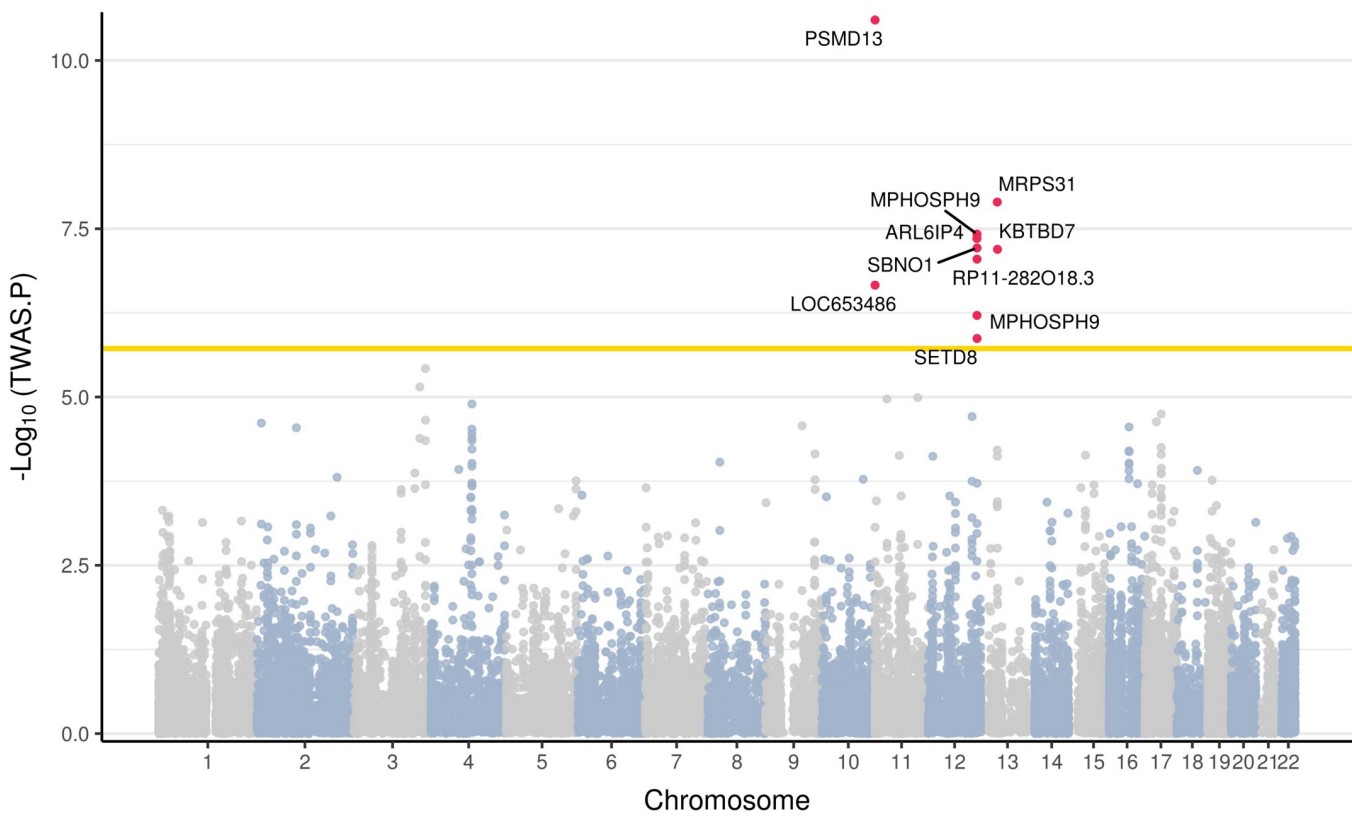

**Fig 1. A Manhattan plot of the TWAS results for UF.** Each dot corresponds to the gene of which predicted expression is associated with UF. The X-axis denotes the chromosome numbers of the genes and the Y-axis denotes $-\log_{10}$(TWAS P-values). The yellow line indicates a Bonferroni significant threshold ($P < 1.90 \times 10^{-6}$). Nine significant TWAS genes are represented as red dots with gene labels.

Then, we performed additional post-GWAS annotation analyses using the SNP2GENE and the MAGMA, which are position-based gene-mapping methods, to confirm the robustness of our TWAS results and validate the novel association from the FUSION. We investigated

**Table 1. A list of significant TWAS genes associated with UF ($P < 1.90 \times 10^{-6}$).**

| Gene | Tissue | Chromosome | LeadGWAS rsID | TWAS.Z (FUSION) | TWAS.P (FUSION) | SNP2GENE | MAGMA |
|---|---|---|---|---|---|---|---|
| *LOC653486* | NTR blood | 11 | rs532483 | 5.1836 | 2.18E-07 | No | No |
| **PSMD13** | NTR blood | | | 6.6731 | 2.50E-11 | Yes | No |
| *RP11-282O18.3*\* | Uterus | 12 | rs641760 | 5.347 | 8.94E-08 | No | No |
| *MPHOSPH9* | Whole blood | | | -5.5 | 3.80E-08 | Yes | Yes |
| | YFS blood | | | -4.9871 | 6.13E-08 | | |
| *SBNO1* | YFS blood | | | 5.416 | 6.10E-08 | Yes | No |
| *ARL6IP4* | Whole blood | | | 5.4736 | 4.41E-08 | Yes | No |
| *SETD8* | Whole blood | | | -4.8328 | 1.35E-06 | Yes | No |
| **KBTBD7**\* | Whole blood | 13 | rs4943810 | -5.4073 | 6.40E-08 | No | No |
| *MRPS31* | YFS blood | | rs7986407 | -5.6906 | 1.27E-08 | Yes | No |

The significant TWAS genes from the FUSION that were neither reported in SNP2GENE nor MAGMA are highlighted in bold. The LeadGWAS rsID indicates the lead SNPs of the locus where each significant TWAS gene is located, and they were calculated by FUSION.

\*Genes that have not been identified in previous UF-related studies

whether the nine susceptibility genes identified by the FUSION overlapped with those identified by the SNP2GENE and/or the MAGMA (S2B Fig and S2 Table). Among the nine TWAS genes from the FUSION, six genes (*PSMD13*, *MPHOSPH9*, *SBNO1*, *ARL6IP4*, *SETD8*, and *MRPS31*) overlapped with genes from the SNP2GENE, while *MPHOSPH9* also overlapped with the genes from the MAGMA. Three genes (*LOC653486*, *RP11-282O18.3*, and *KBTBD7*) were only detected by the FUSION, not from other gene-prioritizing tests (S3 Fig), which suggests that these three genes cannot be detected by conventional methods such as MAGMA or SNP2GENE. Among the three genes only detected by the FUSION, *RP11-282O18.3* and *KBTBD7* were reported for the first time as susceptibility genes for UF, to the best of our knowledge. Colocalization tests were performed to confirm the robustness of the possible causal relationship between the significant TWAS signals and UF. PPs for each TWAS signal associated with UF were calculated by COLOC. The COLOC results showed that six out of the nine significant TWAS genes (*PSMD13*, *MPHOSPH9*, *SBNO1*, *ARL6IP4*, *SETD8*, and *MRPS31*) were replicated in colocalization analyses (PP3+PP4 > 0.8 and PP4/PP3 > 2) (S3 Table and S4 Fig). Together, we identified a total of nine significant TWAS genes including two novel genes—*RP11-282O18.3* and *KBTBD7*—and confirmed the robustness of our results.

## Assessing independence of TWAS signals through conditional and joint analysis

Conditional and joint analyses were applied to genomic regions at chromosomes 11, 12, and 13—as listed in Table 1—to determine whether the expressions of the multiple associated genes in the regions were regulated by the same causal variants. The analyses were carried out for each tissue separately, and the results at the same locus are displayed together in a single plot for better visibility. In the region harboring rs532483 at chromosome 11, GWAS signals showed significant drops after being conditioned on the predicted expression levels of *LOC653486* and *PSMD13* from the NTR blood panel (Fig 2A). Both *LOC653486* and *PSMD13* were observed as independently significant TWAS genes. In the genomic region within 1 Mb of rs641760, five TWAS genes—*RP11-282O18.3*, *MPHOSPH9*, *SBNO1*, *ARL6IP4*, and *SETD8* —were observed from GTEx uterus, GTEx whole blood, and YFS blood panels. Among these five genes, the GWAS signals were significantly decreased when conditioned on the predicted expression level of *MPHOSPH9* from the GTEx whole blood panel, which indicates that *MPHOSPH9* was a jointly significant gene and responsible for the most signals at the locus (Fig 2B). The other four genes—*RP11-282O18.3*, *SBNO1*, *ARL6IP4*, and *SETD8*—were identified as marginally significant genes that were no longer significant after conditioning on the predicted expression level of *MPHOSPH9* (conditioned P-value of *RP11-282O18.3*, 0.23; *SBNO1*, 0.16; *ARL6IP4*, 0.18; and *SETD8*, 0.20).

In the chromosome 13q14.11 region where rs4943810 and rs7986407 were located, the significant GWAS signals at the locus became no longer significant when conditioned on the predicted expression level of *KBTBD7* from GTEx whole blood panel and *MRPS31* from the YFS blood panel, respectively (Fig 2C). The result indicated that both genes are responsible for the effect size of the GWAS locus where they are located. One of the two novel genes, *KBTBD7*, was a jointly significant TWAS gene, and its expected expression level accounted for most GWAS signals at the locus where *KBTBD7* was observed. Together, we identified that *LOC653486*, *PSMD13*, *MPHOSPH9*, *KBTBD7*, and *MRPS31* of the nine TWAS genes, including a novel gene (*KBTBD7*), were independently significant genes after being conditioned on their predicted expression levels, which suggests that we successfully detected robust TWAS genes for UF.

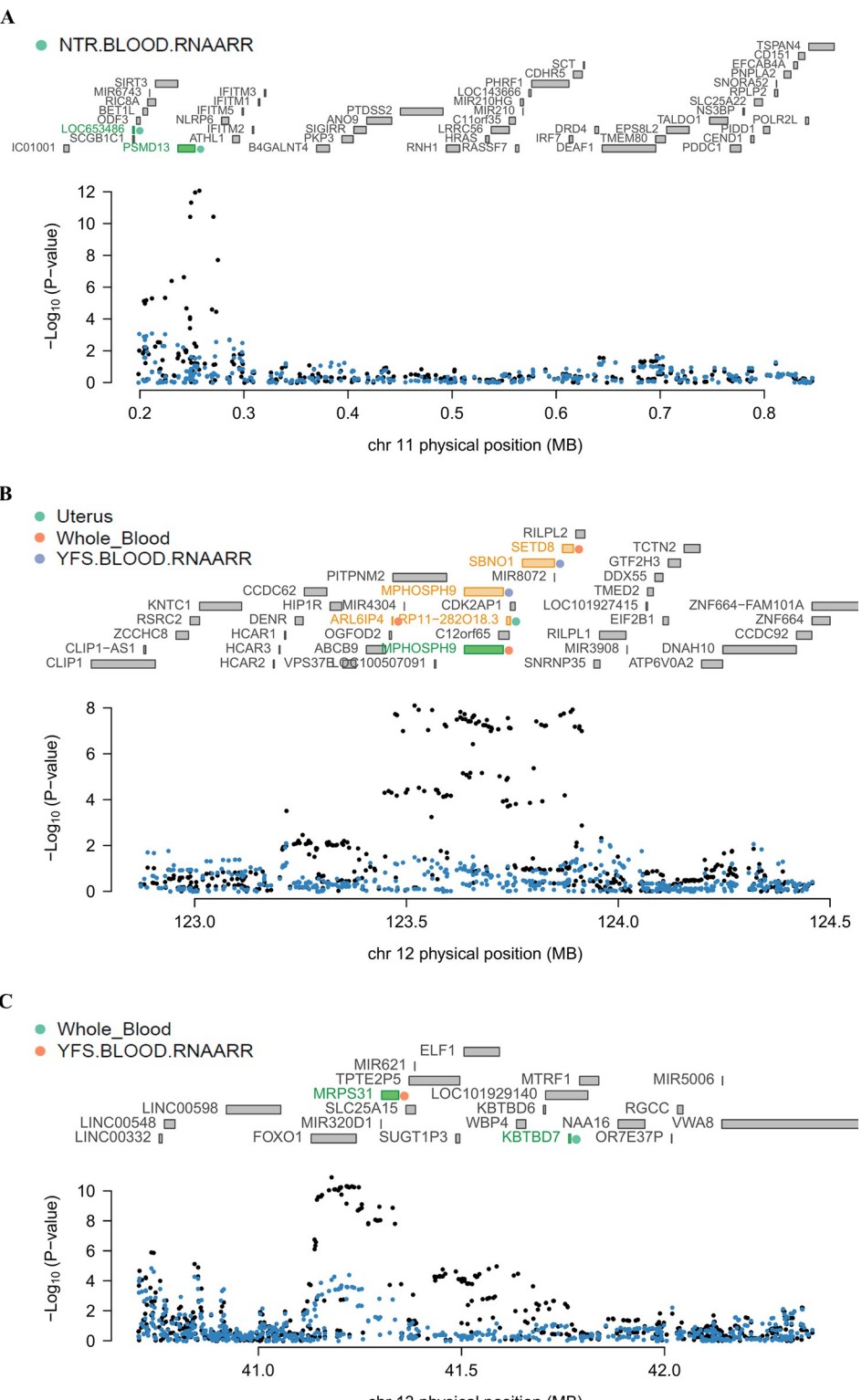

**Fig 2. Regional association plots showing conditional and joint analyses results.** (**A**) The regional association plot of chromosome 11. (**B**) The regional association plot of chromosome 12. (**C**) The regional association plot of chromosome 13. The middle part of each figure represents all genes located in the region. The green bars indicate jointly significant genes that are responsible for most GWAS signals in the region, the yellow bars represent TWAS genes that are no longer significant after accounting for conditionally independent genes, and the gray bars indicate

genes that were neither jointly significant nor marginally significant in that region. The colored dots next to the jointly or marginally significant genes indicate tissue panels where the genes were detected, as summarized in Table 1. The lowest part of each panel is a Manhattan plot of the GWAS signals. Black dots represent the GWAS P-values of SNPs before conditioning tests, and blue dots represent the GWAS P-value of SNPs after removing the effects of jointly significant genes.

### Functional annotation of TWAS genes for UF

In order to explore the biological functions of TWAS genes, we performed GSEA with UF TWAS genes by applying a soft threshold (TWAS P-value < 0.05) using the Metascape annotation tool. A total of 139 pathways were enriched with the 242 positively associated TWAS genes (TWAS Z-score > 0), while 138 biological pathways were enriched with the 235 negatively associated TWAS genes (TWAS Z-score < 0) (S4 and S5 Tables).

To detect representative pathways among the enriched biological pathways, we constructed functional enrichment networks consisting of positively or negatively associated TWAS genes. A functional enrichment network consisting of 139 positively associated pathways was clustered into 14 sub-networks by the MCODE. The connectivity score of each sub-network was calculated (score: 3.00–10.00; median: 7.57) and we defined four clusters that had the top 25% scores as major clusters of the enrichment network (Fig 3A). The four major clusters were categorized into three parental pathways: immune system process, metabolic process, and localization. Next, a total of 15 sub-networks were obtained from the enrichment network of 138 negatively associated pathways and the connectivity scores of sub-networks were calculated (score: 3.33–11.82; median 8.12) by the MCODE. Four clusters with the top 25% scores were defined as major clusters and were classified into four parental pathways: cell cycle, immune system process, mitochondrial gene expression, and cellular component organization or biogenesis (Fig 3B).

### Identification of toxic chemicals associated with UF risk

To identify toxic chemicals such as EDCs that may contribute to the pathogenesis of UF, we performed a chemical–gene interaction analysis using CTD to evaluate the relationship between chemicals and genes. Only six of the nine significant TWAS genes—namely *PSMD13*, *SBNO1*, and *ARL6IP4* that were positively associated with TWAS genes, and *MPHOSPH9*, *KBTBD7*, and *MRPS31* that were negatively associated—were available to search to elucidate their chemical–gene interactions in the CTD. The result showed that a total of 67 chemicals correlated with the six TWAS genes (S6 Table). Among the 67 chemicals, 28 were estimated to increase the expression levels of positively associated TWAS genes (S7 Table), and 33 were estimated to decrease the expression of negatively associated TWAS genes (S8 Table). Notably, the remaining six chemicals were estimated to increase the expression levels of positively associated TWAS genes while also decreasing those of negatively associated TWAS genes. However, we removed valproic acid from the six chemicals because it may either increase or decrease the expression of negatively associated TWAS genes (Table 2). In short, these 66 chemicals may contribute to worsening UF symptoms and/or progression, although the results should be validated by experimental procedures. Among them, aflatoxin B1 and 7,8-dihydro-7,8-dihydroxy benzo(a)pyrene 9,10-oxide (BPDE) are known to act as EDCs [41, 42].

### Discussion

UFs are common benign uterine tumors in women, while their pathological mechanism remains underexplored. Despite previous GWAS contributions to detecting genetic variations associated with UF, only limited insights were provided to explicate the genuine effects of

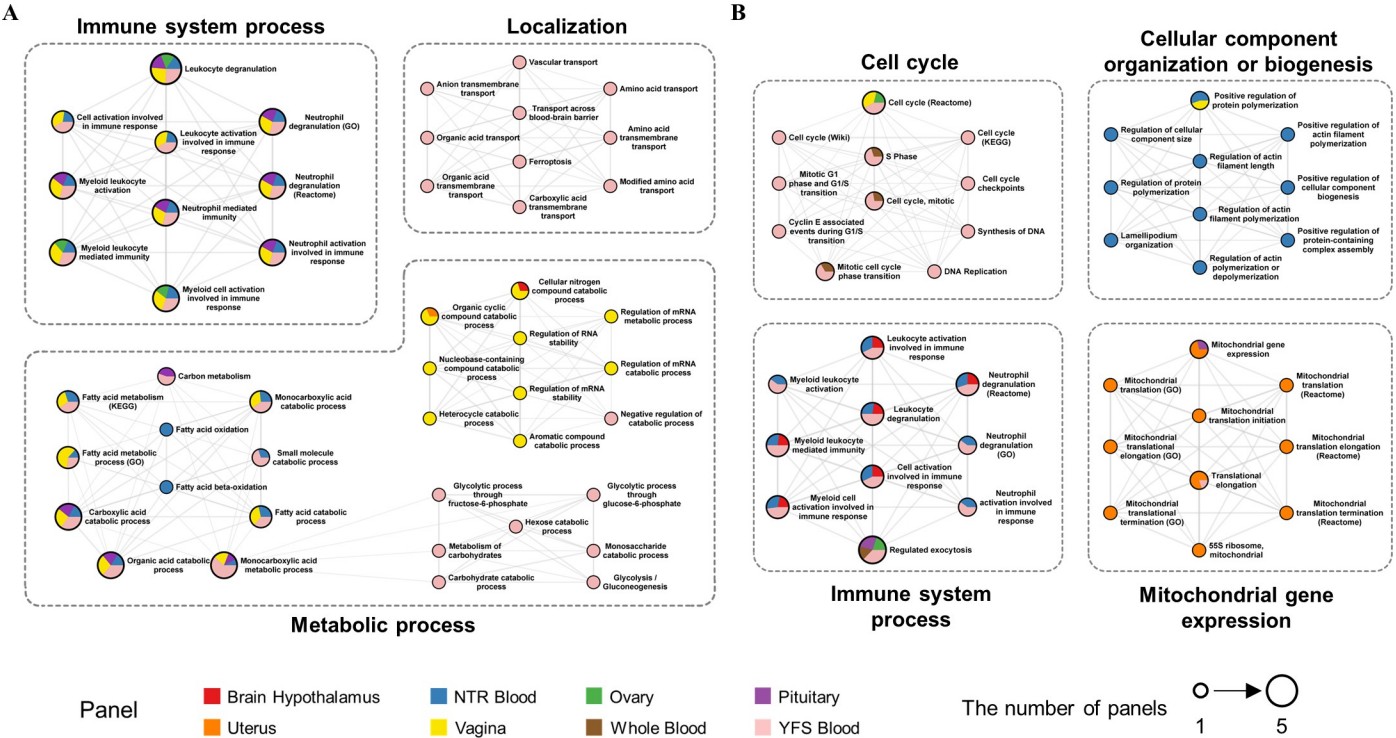

**Fig 3. Functional annotation networks consisting of biological pathways enriched with TWAS genes for UF.** (**A**) Four major clusters grouped into three parental biological pathways were enriched with positively associated TWAS genes (TWAS P-value < 0.05 and Z-score > 0). (**B**) Four major clusters of biological pathways were enriched with negatively associated TWAS genes (TWAS P-value < 0.05 and Z-score < 0). The major clusters are the sub-networks in the top 25% for connectivity score as calculated by the MCODE in each functional annotation network. Each node represented by a pie chart indicates an enriched biological pathway, and the sector size is proportional to the number of genes that originate from each tissue panel. The node size corresponds to the number of panels where TWAS genes were enriched.

genetic risk variants. Here, we integrated the largest UF GWAS data with eight tissue-specific eQTL panels for TI to overcome the limitations of GWAS. Our TWAS successfully identified nine putative causal genes for UF, including two novel genes affected by GWAS SNPs (Fig 1 and Table 1). A previous study that performed a TWAS using Summary-PrediXcan (S-PrediX-can) software with UF GWAS data from 227,329 samples identified *leucine zipper protein 1* from the vagina on chromosome 1 and *oligonucleotide/oligosaccharide binding fold containing 1* on chromosome 10 from the esophagus as potential causal genes for UF [43]. Even though S-PrediXcan reportedly shows results consistent with those of FUSION, our study differs from

**Table 2. A list of five toxic chemicals discovered based on CTD as potentially toxic chemicals for UF.**

| Chemical ID | Chemical name | DrugBank ID | PubChem ID | TWAS genes (Z-scores > 0) | TWAS genes (Z-scores < 0) |
|---|---|---|---|---|---|
| **D019327** | Copper Sulfate | DB06778 | 24462 | *PSMD13, SBNO1* | *KBTBD7, MRPS31* |
| **D016572** | Cyclosporine | DB00091 | 5284373 | *PSMD13* | *KBTBD7* |
| **D004317** | Doxorubicin | DB00997 | 31703 | *PSMD13* | *MPHOSPH9* |
| **D016604** | **Aflatoxin B1** | - | 186907 | *SBNO1* | *MPHOSPH9, KBTBD7* |
| **D015123** | **7,8-Dihydro-7,8-dihydroxybenzo(a)pyrene 9,10-oxide (BPDE)** | - | 53788654 | *ARL6IP4* | *MRPS31* |

The chemicals reported as EDCs are highlighted in bold.

the previous TWAS in a few ways [43, 44]. The previous study utilized S-PrediXcan as the TI method with every tissue-specific panel of GTEx v7 excluding male-specific tissues. We performed TI using FUSION with UF-related tissue panels including two blood panels from NTR and YFS that were not analyzed in the previous publication, because our main focus was potentially strong tissue-specific regulatory effects on the pathogenesis of UF. The reduction in the multiple-testing burden resulting from the use of fewer tissue panels may have contributed to identifying novel putative causal genes for UF that were previously undetected [43].

The fact that the results from several studies were in line with our TWAS results supports the robustness of our TWAS genes. Homologs of *LOC653486* and *PSMD13* were implicated in UF risk loci by a previous GWAS study by Nakamura *et al.* [20]. Seven significant TWAS genes from our study—*LOC653486*, *PSMD13*, *MPHOSPH9*, *SBNO1*, *ARL6IP4*, *SETD8*, and *MRPS31*—had been previously identified as residing in genetic regions associated with UF [45]. These seven genes were also found to be related to immune response, tumorigenesis, or metabolic diseases. One study showed that *LOC653486* is significantly associated with nasal polyposis and asthma, which are chronic inflammatory diseases [46]. Another revealed that *PSMD13* is associated with the number of platelets, which are mediators in the immune and inflammatory response [47]. Two adjacent genes *MPHOSPH9* and *SBNO1*, located in 12q24, were reported as having susceptible associations with type 2 diabetes [48, 49]. A study on the shared risk of schizophrenia and cardiometabolic diseases including obesity, body mass index, and type 2 diabetes suggested that *MPHOSPH9*, *ARL6IP4*, and *SETD8* are pleiotropic risk genes [50]. *SETD8*, a subtype of lysine demethylase, was also studied to examine how its dysregulation is involved in the progression of various biological processes including tumorigenesis [51]. It was found that *MRPS31* encodes ribosomal protein Imogen 38, which is a suggested target for autoimmune attack in type 1 diabetes [52]. In addition, we identified two novel susceptibility genes for UF, *RP11-282O18.3* and *KBTBD7* (Fig 2B and 2C). *RP11-282O18.3* and *KBTBD7* have been mentioned as being connected with immunological features and various diseases. *RP11-282O18.3*, a long non-coding RNA that likely affects non-allergic asthma, has been shown to be involved in the network comprising genes relevant to the estradiol phenotype of polycystic ovary syndrome (PCOS) [53]. Wise *et al.* described that PCOS patients had a higher UF incidence than healthy controls and suggested that PCOS and UF are positively correlated with each other [54]. Studies have revealed that *KBTBD7* encodes a transcriptional activator forming a complex with cullin 3 to regulate the degradation of neurofibromin involved in the Ras/extracellular signal-regulated kinase pathway, playing crucial roles in the development of various malignant tumors in the event of dysfunction [55–57]. *KBTBD7* has been shown to increase the transcription of *activator protein-1* (*AP-1*) and serum response element (SRE) [55]. Increased transcriptions of *AP-1* and SRE have been found to positively regulate the mitogen-activated protein kinase signaling pathway inducing inflammatory responses [55, 58]. Previous studies reported that microRNA-21 (miR-21), which is also observed in humans, induces *Kbtbd7* mRNA degradation and inhibits the translation of *Kbtbd7* in mice [59, 60]. Additionally, miR-21 was more expressed in UF samples than in normal myometrium samples [61]. Since our datasets do not contain data on microRNAs, we cannot be certain whether miR-21 played a role in *KBTBD7* and UF pathogenesis at this point. However, we cannot rule out the possibility that miR-21 plays a major role in our findings that *KBTBD7* is a novel susceptibility gene for UF. It may be worthwhile to perform an *in silico* and experimentation study on human UF patients and healthy control subjects side-by-side to prove our hypothesis. Together, we suggest that our study successfully identified robust novel TWAS genes that are putatively causal for the pathogenesis of UF.

In the functional annotation of TWAS genes, both positively and negatively associated TWAS genes of UF were involved in the immune system process (Fig 3). The clusters of the

immune system process showed high connectivity scores from both enrichment networks composed of positively and negatively associated pathways, both scoring 10.00. TWAS genes from seven out of the eight tissue panels, all except the uterus panel, were enriched in the immune system process. Given this result, we believe that the immune system process pathway participates in the underlying pathogenesis of UF in multiple tissue levels. We identified that positively associated UF TWAS genes were enriched in the metabolic process, which may support the robustness of our functional analysis since metabolic syndromes are well-known risk factors for UF [62]. In addition, mitochondria participate in the central metabolic pathway, which explains the high connectivity score of the mitochondrial gene expression cluster (score: 10.00) in the enrichment networks of negatively associated pathways as well as the metabolic process cluster [63]. The metabolic process pathway was mainly enriched in the blood tissue panels, whereas the mitochondrial gene expression pathway was mostly detected in the uterus panel. It has been reported that the differential expression of mitochondrial progesterone receptors is associated with UF, since progesterone may affect the growth of UF by altering mitochondrial activity [64]. It is possible that the genetic variants of UF have tissue-specific effects on metabolic processes and mitochondrial gene expression pathways. Some of the other TWAS genes positively associated with UF were found to be enriched in pathways belonging to the parental pathway 'localization'. A previous study reported that exposure to nonylphenol and di(2-ethylhexyl) phthalate modified the localization and colocalization patterns of uterine estrogen receptors and progesterone receptors, resulting in changes in the proliferation patterns of endometrial tissues [65]. We believe that the localization of cellular molecules, especially steroid receptors, may affect cell proliferation and induce the formation of UF. The localization cluster was only enriched in YFS blood panels, suggesting that their interference with the localization of intracellular material in the blood cells could be related to UF. The cell cycle cluster showed the highest connectivity score (score: 11.82) among the four major clusters enriched with negatively associated TWAS genes. Since the loss of cell cycle regulation is a critical characteristic of tumor progression, our GSEA results also explain why some UF patients develop tumors [66]. Finally, we identified the cellular component organization or biogenesis cluster, which has rarely been referenced in UF pathogenesis. Taken together, we found the immune system process, metabolic process, mitochondrial gene expression, localization, cell cycle, and cellular component organization as the key pathways that may be related to the pathophysiology of UF.

Using the CTD, we identified five toxic chemicals associated with our TWAS genes expected to be involved in the pathogenesis of UF (Table 2). The five chemicals comprising two drugs (cyclosporine and doxorubicin) and three chemicals (copper sulfate, aflatoxin B1, and BPDE), which enhance the expression levels of both positively and negatively associated TWAS genes, were previously implicated in female reproductive diseases such as endometrial cancer, ovarian cancer, and PCOS. Among these potential chemical hazards, copper sulfate had associations with four TWAS genes, implicating its detrimental effects on UF. Copper has been found to play an important role in tumor growth by promoting tumor angiogenesis and stimulating cell proliferation [67, 68]. A previous study showed that the detected level of serum copper was higher in women with hysteromyoma than in healthy controls and suggested that copper is interrelated with UF, known as a hysteromyoma disease [69]. The two drugs—cyclosporine and doxorubicin—have been indirectly associated with UFs in terms of female-specific diseases. Cyclosporine, used as an immunosuppressant, reportedly promotes tumor angiogenesis and causes fibroadenoma, which is positively associated with the pathogenesis of UF [70–72]. Doxorubicin is a treatment for uterine sarcoma but has cardiotoxicity that accelerates the risk of cardiovascular disease in some female breast cancer patients [73–75]. Cardiovascular risk factors are more prevalent in UF patients than in controls and it was suggested that there are common risk factors between cardiovascular disease and UF such as BMI and hypertension [76]. Thus,

the use of these two drugs may indirectly increase the risk of UF. The other two chemicals—aflatoxin B1 and BPDE, which have been described as human carcinogens in previous studies—are regarded as potentially toxic chemicals and EDCs [41, 42, 77–80]. Aflatoxin B1 was reported to induce uterine damage in mice and estrogen synthesis by changing physiological aromatase functions, causing endocrine disruptors in the placenta [41, 77]. BPDE was revealed to be a benzo(a)pyrene metabolite that causes toxicity in various organs [78]. Exposure to benzo(a)pyrene reportedly affected the occurrence of infertility and ovarian cancer and increased the prevalence of UF in a female genital tract study [78, 81, 82]. Benzo(a)pyrene has also been shown to be a xenoestrogen that may affect the growth of UF by mimicking the effects of estrogen and acting as an estrogen receptor agonist in rat uterine leiomyoma cells [83–85]. BPDE treated in mice ovaries is also known to affect the suppression of steroidogenic enzymes and induce ovarian disorders [86]. Our data suggest that exposure to one or more of these chemicals may contribute to the occurrence of UF. Aflatoxin B1, known as a secondary fungal metabolite, is widely found in foods such as rotten nuts or dried fruits [87]. High concentrations of benzo(a)pyrene are found in heat-treated foods such as charcoal-grilled meat [88]. Based on our results, women with certain genetic backgrounds of UF combination with chronic dietary exposure to aflatoxin B1 and/or BPDE may show a higher risk of UF; although the actual effects of the exposure should be validated through experimental studies, it is also true that exposure to aflatoxin B1 and BPDE should always be avoided because they are toxic. Overall, our results suggest that these five toxic chemicals may increase the development of UF and their intake needs to be carefully monitored due to their various side effects.

Although this study contributed to comprehending the genetic and chemical risks associated with UF, several limitations remain to be addressed. Our TWAS was conducted on autosomes because the FUSION only implements sumstats-file formatting of GWAS data on autosomes, even though rearrangements of the X chromosome were reportedly implicated in UF [38]. Since complicated biological phenomena at the X chromosome such as mosaic inactivation may play a crucial role in UF pathogenesis, further studies on sex chromosomes are also warranted to investigate their overall genetic influences on UF etiologies when the technology becomes available. The GTEx panels contain a significant proportion of samples from women aged 50–70 years who are post-menopausal, whereas the majority of cases of UFs occur in premenopausal women. As the tissue panels used in our study did not consist only of age groups with a high UF risk, age-related statistical specificity was reduced. In addition, since the panels used in this study were single-tissue eQTL panels, the statistical power may have been insufficient to detect all true associations. This lack of resources may have contributed to the discrepancy between the number of significant loci in our study and previous UF GWASs. Although it is difficult to address this issue immediately due to a lack of resources, increasing the sample size of UF-related tissue panels or publication of robust multi-content panels may allow the detection of more genotype–gene expression associations for UFs. Even though we identified significant TWAS genes as potential causal genes of UF and confirmed the robustness of this result through the colocalization tests, three genes including our novel findings, *RP11-282O18.3* and *KBTBD7*, were not replicated in the colocalization results. Therefore, the actual effects of our significant TWAS genes on UF risk should be validated through experimental studies. We detected five toxic chemicals that may increase the pathogenesis of UF; however, their effects under physiological conditions should be validated since the results were obtained using *in silico* analyses. Despite these limitations, we believe that this study successfully identified TWAS genes associated with UF risks and potentially toxic chemicals expected to influence TWAS genes, which suggests that our results may contribute to a deeper understanding of UF etiologies and provide informative notifications of potentially risky chemicals associated with UF.

## Supporting information

**S1 Fig. Circos plots showing how marginally significant TWAS gene lists (P < 0.05) for each of the eight eQTL panels overlap.** On the outside, the arc represents the eight eQTL panels. On the inside, the dark orange arc represents genes shared by several panels and the light orange arc represents genes unique to those panels. The purple lines link the genes shared by several panels. (**A**) The plot shows the shared genes between positively associated TWAS genes (TWAS Z-score > 0). (**B**) The plot shows shared genes between negatively associated TWAS genes (TWAS Z-score < 0).
(TIF)

**S2 Fig. Manhattan plots of the UF GWAS summary statistics data analyzed and visualized by FUMA.** (**A**) A Manhattan plot of the input GWAS summary statistics. (**B**) A Manhattan plot of the MAGMA results. Significant prioritized genes associated with SNPs are visualized with gene symbols. The dashed red line indicates a Bonferroni significant threshold (P < 1.90 × 10^{-6}).
(TIF)

**S3 Fig. A Venn diagram showing the overlap between UF-associated genes discovered by TWAS analysis, SNP2GENE process, and MAGMA.** The number of genes only identified in the FUSION is highlighted in bold.
(TIF)

**S4 Fig. A ternary plot of colocalization test results.** PP0–4 indicate PPs of five hypotheses (H0–4). The gray dots are the genes that were not significant in either TWAS or the colocalization tests. The red and blue dots indicate the significantly associated genes in TWAS and the colocalization tests, respectively. The genes that were prioritized in both TWAS and the colocalization tests are represented as purple dots.
(TIFF)

**S1 Table. Every TWAS association from the eight tissue panels.**
(XLSX)

**S2 Table. UF-associated genes discovered by the FUMA-SNP2GENE process.**
(XLSX)

**S3 Table. Colocalization test results of COLOC-prioritized and significant TWAS genes.**
(XLSX)

**S4 Table. Gene sets enriched with positively associated TWAS genes provided by the Metascape.**
(XLSX)

**S5 Table. Gene sets enriched with negatively associated TWAS genes provided by the Metascape.**
(XLSX)

**S6 Table. Chemicals correlated with the significant TWAS genes of UF.**
(XLSX)

**S7 Table. Chemicals that increase the expression levels of significant TWAS genes positively associated with UF.**
(XLSX)

**S8 Table. Chemicals that decrease the expression levels of significant TWAS genes negatively associated with UF.**
(XLSX)

## Acknowledgments

The authors appreciate the researchers who deposited their data in public database.

## Author Contributions

**Conceptualization:** Gayeon Kim, Gyuyeon Jang, Jaeseung Song.

**Data curation:** Gayeon Kim, Gyuyeon Jang.

**Formal analysis:** Gayeon Kim, Gyuyeon Jang, Jaeseung Song, Daeun Kim, Sora Lee.

**Investigation:** Gayeon Kim, Gyuyeon Jang, Jaeseung Song.

**Methodology:** Gayeon Kim, Gyuyeon Jang, Jaeseung Song.

**Project administration:** Jaeseung Song, Jong Wha J. Joo, Wonhee Jang.

**Resources:** Jong Wha J. Joo, Wonhee Jang.

**Software:** Gayeon Kim, Gyuyeon Jang, Jaeseung Song, Daeun Kim, Sora Lee.

**Supervision:** Jong Wha J. Joo, Wonhee Jang.

**Validation:** Gayeon Kim, Gyuyeon Jang, Jaeseung Song, Daeun Kim.

**Visualization:** Gayeon Kim, Gyuyeon Jang.

**Writing – original draft:** Gayeon Kim, Gyuyeon Jang, Jaeseung Song, Daeun Kim.

**Writing – review & editing:** Gayeon Kim, Gyuyeon Jang, Wonhee Jang.

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
