## [Decision Letter · Decision Letter 0]

9 Jun 2022

PONE-D-22-12765A transcriptome-wide association study of uterine fibroids to identify potential genetic markers and toxic chemicalsPLOS ONE

Dear Dr. Jang,

Thank you for submitting your manuscript to PLOS ONE. After careful consideration, we feel that it has merit but does not fully meet PLOS ONE’s publication criteria as it currently stands. Therefore, we invite you to submit a revised version of the manuscript that addresses the points raised during the review process. Several important issues were raised by Reviewer 2 and the authors should address all their comments to improve the quality of the study. Particular attention should be paid to comment #9: LD contamination is a major issue for TWAS and leads to spurious associations. Colocalisation analysis should be performed using the corresponding eQTL data to prioritise for causal TWAS associations.

We look forward to receiving your revised manuscript.

Kind regards,

Dylan Glubb

Academic Editor

PLOS ONE

Journal Requirements:

"This work was supported by the Dongguk University Research Fund of 2020 and 2021 (S-2021-G0001-00094)."

Reviewers' comments:

Reviewer's Responses to Questions

**Comments to the Author**

1. Is the manuscript technically sound, and do the data support the conclusions?

Reviewer #1: Yes

Reviewer #2: Partly

2. Has the statistical analysis been performed appropriately and rigorously? 

Reviewer #1: Yes

Reviewer #2: Yes

3. Have the authors made all data underlying the findings in their manuscript fully available?

Reviewer #1: Yes

Reviewer #2: Yes

4. Is the manuscript presented in an intelligible fashion and written in standard English?

Reviewer #1: Yes

Reviewer #2: Yes

5. Review Comments to the Author

Reviewer #1: The authors compile together publicly available GWAS summary statistics with other publicly available sources of gene expression data. The study is driven by material derived from public databases and readily available in silico data-analysis methods. Overall, their choice of methods appear reasonable, and the presentation is adequate. The material has several limitations, some of which are already mentioned in the discussion.

Minor comments:

i) Discuss the limitations of the material used: first, the age distribution of the tissue material does not appear suitable, simply because GTEx tissues originate predominately from 50-70 year-old individuals (postmenopausal individuals), while uterine fibroids typically occur premenopausal. Second, discuss limitations of statistical power (see also my next comment).

ii) Discuss the discrepancy between the large number of significant GWAS loci and small number of significant TWAS loci. The former is more than 20 loci (according to Supplementary Fig S4 and many previous UF GWAS studies), while the latter, your novel results, originate from just three distinct loci. - Is it the lack of power in the GTEx material which explains that majority of the GWAS hits do not show TWAS?

Reviewer #2: The manuscript by Kim et al describes transcriptome-wide analyses of uterine fibroid risk with extensions to chemical target identification. The work is interesting and leverages many publicly available resources. The results are interesting, however, the organization needs some adjustment. Mainly there are several paragraphs which would be better housed in the discussion, which would then likely need to be shortened considerably. There are also some issues of clarity throughout.

Specific Comments:

1. Lines 48-58: The 2nd paragraph of the introduction mentions genetic risk factors, but then proceeds to predominantly discuss cytogenetic changes observed in fibroid tumors, which are not necessarily risk factors as they can only be observed once a tumor has formed. It may be better to separate the heritability concept from the cytogenetic alterations and instead place it with the germline genetic analyses which are actually evaluating disease risk.

2. Lines 69-72: The GWAS paragraph includes mention of one GWAS (out of at least 9 that have been performed) and one admixture mapping study, neither of which contain the results which were used in the present manuscript.

3. Line 81: The Gallagher et al paper that produced the GWAS results investigated in this paper but which is not cited directly is not the most recent GWAS.

4. Line 115: it is unclear whether the 26,316 represents the total number of unique genes across all 8 tissues, or the sum of genes predicted in each tissue (the technical number of actual tests)

5. Are the joint and conditional tests of TWAS loci performed within FUSION or was there another software/analysis package used for these steps?

6. Line 152: What does CTD stand for?

7. The chemical-gene interaction analysis methods need a bit of rewording to make a couple of items clear. Line 155: chemicals that may induce development of fibroids. Lines 159-160: “the expression caused by the UF” – the gene expressions you have predicted are based on risk OF developing a fibroid, and the tissues are not all from uterus, let alone from the fibroid itself, so this needs to be reworded to be more accurate.

8. The results should indicate that only one of the nine genes was significant in uterus, and that all of the rest were significant in only blood (what is the rationale for including three different whole blood tissue sets? The results do not seem consistent between the three).

9. It doesn’t appear that any colocalization methods were used to eliminate the potential for LD contamination driving the observed associations in TWAS. This should be considered for robustness.

10. The paragraph beginning on line 203 should be moved to discussion, it is not presentation of results.

11. How does the conditional analysis handle the differing tissues from which the expression is associated? How were the mentioned SNPs (lines 231, 234, 257) selected? Lead SNPs in the region from FUMA, or known eQTLs for the genes of interest?

12. Numbers should be included for the numbers of genes included in the up- and down-regulated subsets (lines 273 and 274)

13. A large portion of the paragraph beginning on line 299 should be moved to discussion.

14. The statement on line 346 is overstated in the absence of experimental or observational data.

15. The paragraphs beginning on line 352 and line 360 seem more like discussion.

16. The tissues from which the significant results from the previous study’s TWAS came from should be noted.

17. A 7% sample size increase at the scale of 200 thousand participants seems like a stretch to claim increased power for detection. It is notable that the 5 of the 10 significant results came from tissue sources not analyzed in the previous publication, and likely that the differing software and reduced multiple-testing burden in your paper also contributed to the differences in findings.

18. The sentence on lines 426-427 doesn’t make sense.

19. The paragraph beginning on line 436 has 10 lines about a chemical you excluded from results, this seems unnecessary to interpretation of the current results.

20. The sentence on lines 464-465 recommending that women with certain genetic profiles should avoid dietary chemical exposures is a complete overstatement. At MOST this should be presented as a relationship which should be evaluated in model organism experimental or human observational studies well before guidelines should be presented to patients, who most likely do not know what the risk “genetic backgrounds of UF” are, let alone whether they carry such profiles.

6. PLOS authors have the option to publish the peer review history of their article (what does this mean?). If published, this will include your full peer review and any attached files.

Reviewer #1: No

Reviewer #2: No

---

## [Author Response · Author response to Decision Letter 0]

5 Aug 2022

We appreciate all of the helpful comments about our work. Detailed responses and cover letters are provided in the attached file.

---

## [Decision Letter · Decision Letter 1]

7 Sep 2022

A transcriptome-wide association study of uterine fibroids to identify potential genetic markers and toxic chemicals

PONE-D-22-12765R1

Dear Dr. Jang,

We’re pleased to inform you that your manuscript has been judged scientifically suitable for publication and will be formally accepted for publication once it meets all outstanding technical requirements.

Kind regards,

Dylan Glubb

Academic Editor

PLOS ONE

Additional Editor Comments (optional):

The manuscript is provisionally accepted but the authors need to revise the following sentence (as per Reviewer 2's comments): "Based on our results, women with certain genetic backgrounds of UF due to chronic dietary exposure to aflatoxin B1 and/or BPDE may show a higher risk of UF;". As the reviewer states "the genetic backgrounds of UF are likely not due to the exposure, but rather I believe the authors are trying to state that those genetic backgrounds in combination with the exposure may increase risk" and thus the sentence should be revised accordingly.

Reviewers' comments:

Reviewer's Responses to Questions

**Comments to the Author**

1. If the authors have adequately addressed your comments raised in a previous round of review and you feel that this manuscript is now acceptable for publication, you may indicate that here to bypass the “Comments to the Author” section, enter your conflict of interest statement in the “Confidential to Editor” section, and submit your "Accept" recommendation.

Reviewer #1: All comments have been addressed

Reviewer #2: All comments have been addressed

2. Is the manuscript technically sound, and do the data support the conclusions?

Reviewer #1: Yes

Reviewer #2: Yes

3. Has the statistical analysis been performed appropriately and rigorously? 

Reviewer #1: Yes

Reviewer #2: Yes

4. Have the authors made all data underlying the findings in their manuscript fully available?

Reviewer #1: Yes

Reviewer #2: Yes

5. Is the manuscript presented in an intelligible fashion and written in standard English?

Reviewer #1: Yes

Reviewer #2: Yes

6. Review Comments to the Author

Reviewer #1: (No Response)

Reviewer #2: The authors have adequately addressed all of my comments. My one very minor remaining comment is related to the revised sentence in the discussion "Based on our results, women with certain genetic

backgrounds of UF due to chronic dietary exposure to aflatoxin B1 and/or BPDE may show a

higher risk of UF;". The genetic backgrounds of UF are likely not due to the exposure, but rather I believe the authors are trying to state that those genetic backgrounds in combination with the exposure may increase risk

7. PLOS authors have the option to publish the peer review history of their article (what does this mean?). If published, this will include your full peer review and any attached files.

Reviewer #1: No

Reviewer #2: No

---

## [Editor Report · Acceptance letter]

20 Sep 2022

PONE-D-22-12765R1 

A transcriptome-wide association study of uterine fibroids to identify potential genetic markers and toxic chemicals 

Dear Dr. Jang:

I'm pleased to inform you that your manuscript has been deemed suitable for publication in PLOS ONE. Congratulations! Your manuscript is now with our production department. 

Kind regards, 

on behalf of

Dr. Dylan Glubb 

Academic Editor

PLOS ONE